# Oxidative Stress and Beta Amyloid in Alzheimer’s Disease. Which Comes First: The Chicken or the Egg?

**DOI:** 10.3390/antiox10091479

**Published:** 2021-09-16

**Authors:** Elena Tamagno, Michela Guglielmotto, Valeria Vasciaveo, Massimo Tabaton

**Affiliations:** 1Department of Neuroscience, University of Torino, Via Cherasco 15, 10126 Torino, Italy; michela.guglielmotto@unito.it (M.G.); valeria.vasciaveo@edu.unito.it (V.V.); 2Neuroscience Institute of Cavalieri Ottolenghi Foundation (NICO), University of Torino, Regione Gonzole 10, Orbassano, 10043 Torino, Italy; 3Unit of Geriatric Medicine, Department of Internal Medicine and Medical Specialties (DIMI), University of Genova, Viale Benedetto XV 6, 16132 Genova, Italy; mtabaton@neurologia.unige.it

**Keywords:** oxidative stress, β amyloid, Alzheimer’s disease

## Abstract

The pathogenesis of Alzheimer’s disease involves β amyloid (Aβ) accumulation known to induce synaptic dysfunction and neurodegeneration. The brain’s vulnerability to oxidative stress (OS) is considered a crucial detrimental factor in Alzheimer’s disease. OS and Aβ are linked to each other because Aβ induces OS, and OS increases the Aβ deposition. Thus, the answer to the question “which comes first: the chicken or the egg?” remains extremely difficult. In any case, the evidence for the primary occurrence of oxidative stress in AD is attractive. Thus, evidence indicates that a long period of gradual oxidative damage accumulation precedes and results in the appearance of clinical and pathological AD symptoms, including Aβ deposition, neurofibrillary tangle formation, metabolic dysfunction, and cognitive decline. Moreover, oxidative stress plays a crucial role in the pathogenesis of many risk factors for AD. Alzheimer’s disease begins many years before its symptoms, and antioxidant treatment can be an important therapeutic target for attacking the disease.

## 1. Introduction

Alzheimer’s disease (AD) is considered the leading cause of dementia and is becoming one of the most expensive and deadly diseases of our time [1]. Thus, it is estimated that 50 million people worldwide endure dementia, and this number is set to rise to 152 million in 2050 [2,3]. Moreover, Alzheimer’s patients need specialized and expensive care, the annual cost of treatment worldwide is around a trillion US dollars, and it is predicted that this cost will significantly increase by 2030 [4].

The pathophysiology of the disease is complex and multifactorial and certainly not entirely known [3]. There are two markers of the disease. One is β amyloid (Aβ), which accumulates abnormally in AD brain tissues and forms extracellular plaques known to induce synaptic alterations and neurodegeneration [5,6]. The other is Tau protein, which forms intracellular neurofibrillary tangles that are also responsible for neurodegeneration [7,8].

AD is traditionally divided into two forms: early and late onset forms. The early onset form is closely associated with mutations on three genes: the genes that encode for the amyloid precursor (APP) and for presenilin 1 and presenilin 2, which represent the catalytic core of γ-secretase, a key enzyme for the production of Aβ [9]. The late onset form is caused by complex interactions between genetic and environmental factors [10,11]. APOE is the gene that was first associated with the development of the late form [12] compared to other predisposing genes that have been described. These include genes involved in neuroinflammation (such as TREM2, TYROBP, and CD33) [13,14,15], memory (CR1, PICALM, and BIN1) [16,17,18], and lipid metabolism (ABCA7 and CLU) [19,20].

Thus, although there are numerous factors associated with the development of the disease, Aβ represents one that is the most closely related to its pathogenesis. Aβ is composed of polypeptides of various lengths, and most of those found in the plaques are 40 and 42 amino acids long [6]. Aβ 40 is the most abundant form and accounts for 90% of the amyloid present in the plaques; however, the longer form (Aβ 42) is predominant in the initial phases and aggregates faster than Aβ 40 [21]. Aβ is produced from the amyloid precursor protein (APP). APP is an integral membrane protein with a large, extracellular N-terminus and a shorter, cytoplasmic C-terminus [22,23]. The amyloidogenic processing of APP involves two sequential cleavages operated by the β-secretases and γ-secretases. The β-secretase (BACE1) cleaves APP, generating an extracellular soluble fragment called sβAPP and an intracellular C-terminal end termed C99 [24]. C99 is further cleaved, within the membrane, by the γ-secretase [25]. The γ-cleavage produces Aβ fragments of different lengths, and these are predominantly Aβ 40 and Aβ 42 [25].

In the last 30 years, many authors supported the hypothesis of the “amyloid cascade” that considers Aβ to possess a crucial role in the pathogenesis of the disease [26,27,28,29,30]. According to this theory, the accumulation and deposition of Aβ is responsible for Tau aggregation and, therefore, for the cognitive and mnemonic decline observed in AD patients [31].

However, neither the rate of dementia nor the extent of neurological damage is correlated with the Aβ aggregates [32]. Studies on transgenic mice carrying mutations in the gene for APP demonstrated the existence of soluble Aβ (sAβ) oligomers long before the deposition of β-amyloid, further supporting the hypothesis that, in particular conditions such as over-production, soluble Aβ aggregates exist in the human brain even in the absence of plaques. The term soluble Aβ is a working definition, which combines all forms of Aβ derived from physiological brain extracts that remain in aqueous solution after high-speed centrifugation [33]. Soluble oligomers released by cells have been shown to cause neuronal dysfunction in vivo as well as synaptic loss and cognitive impairment [34,35,36,37,38,39,40].

Studies on Down Syndrome (DS) demonstrate that the presence of the APP gene in triplicate is most likely the cause of the early onset of signs of dementia in people with DS, see review [34]. Moreover, the accumulation of Aβ is thought to play a fundamental role in triggering synaptic dysfunction in neurons and results in their eventual loss. Experimentally, soluble Aβ oligomers have been found to selectively block hippocampal long-term potentiation (LTP), widely believed to underlie learning and memory [35,36]. Importantly, active and passive Aβ immunotherapies have been shown to protect against the neuropathological and cognitive deficits observed in APP transgenic models of AD and in AD patients [37,38].

Furthermore, the existence of a relationship between the molecular model of sAβ and the phenotype of the disease was suspected, and this connection reflected the type and degree of neurotoxicity of Aβ. The species of sAβ associated with physiological aging of the brain were compared with those present in sporadic cases of Alzheimer’s disease, and it was found that the species of sAβ in the two different conditions are different [39]. However, it must be emphasized that a recent paper has not found differences in homomeric, heteromeric and soluble Aβ between normal and Ad brains [40].

It has also been observed by Tagliavini’s group that the assembly of mixtures of Aβ peptides into different Aβ seeds results in the formation of different subtypes of amyloid with distinctive physicochemical and biological properties resulting in the generation of distinct AD molecular subgroups [41]. Selkoe’s group suggested that preventing soluble Aβ oligomer formation and targeting their N-terminal residues with antibodies could be an attractive combined therapeutic approach [42].

Thus, there is a balance between Aβ synthesis and degradation through self-regulatory pathways [43]. However, the destruction of this equilibrium results in the overproduction of Aβ in brain tissue [6]. One of the main mechanisms that break this balance is oxidative stress (OS) and neuroinflammation [44]. OS is caused by a derangement between the production of reactive oxygen species (ROS) and the antioxidant defences of an organism.

OS and Aβ are linked to each other because Aβ induces OS in vivo and in vitro [44,45,46], and OS increases the production of Aβ [47,48,49,50].

## 2. Aβ vs. Oxidative stress

There are numerous mechanisms described in the literature by which beta amyloid (Aβ) mediates oxidative damage (Figure 1).

### 2.1. Mitochondria

Aβ interferes with the normal mitochondrial activity, causing dysfunction that results in oxidative stress [51]. Thus, neurons are cells that require high energy levels to perform numerous functions, such as the generation of action potentials, nerve transmission, and axonal transport [52].

The alteration of oxidative phosphorylation (OXPHOS) involves a reduction in the efficiency to transfer electrons, which in turn results in an increase in ROS production predominantly at level of complex I and complex III. These ROSs generated at the chain level unfold their damaging action mainly on mitochondrial macromolecules [53]. The peptide Aβ not only promotes the generation of ROS at the level of the mitochondria, but inhibits, at the same time, ROS removal. In fact, it has been shown that Aβ is able to inhibit mitochondrial superoxide dismutase (MnSOD), the enzyme most involved in the detoxification of the anion superoxide and protection from peroxidative damage [54,55]. Aβ is also capable of binding and inhibiting mitochondrial alcohol dehydrogenase known as ABAD (Aβ binding alcoholdehydrogenase). ABAD has a protective role, being responsible for the detoxification from aldehydes such as 4-hydroxinonenal. The interaction between the Aβ peptide and ABAD compromises the detoxification process for which the enzyme is responsible for, causing lipid peroxidation, ROS generation, and mitochondrial dysfunction [56].

Furthermore, from the direct impact of Aβ on mitochondria, some authors also demonstrated that mitochondrial DNA is altered in elderly and AD patients [57]. Various factors can influence mitochondrial activity, contributing to AD progression. [58]. In brain tissue of AD cases, there is a downregulation of genes in mitochondrial complex I of the OXPHOS [59], and OS is implicated in mtDNA damage [60].

### 2.2. Transition Metals

Another mechanism found dysregulated in AD is the homeostasis of metals such as iron (Fe), copper (Cu), and zinc (Zn). The blood–brain barrier tightly regulates the concentration of these metals, but their levels significantly increased in AD patients [61,62]. Studies have shown that Aβ plaques contain traces of these metals [63], and Aβ can reduce Fe (III) or Cu (II) to induce hydrogen peroxide (H_2_O_2_) production, contributing to OS in AD [64]. Moreover, some studies suggested that these metals can increase Aβ polymerization; thus, neuroblastoma cells treated with Fe^3+^ caused an increase in BACE1 activity that, in turn, promotes Aβ production [65]. Moreover, zinc is also abundant in amyloid plaques, suggesting its role in AD. Zinc interacts with Aβ protein, and this interaction can be prevented by chelators [66,67]. ZnAβ oligomers mediate stronger toxicity than amyloid-beta derived diffusible ligands (ADDLs) by cell viability assays [68]. The ex vivo study showed that ZnAβ oligomers inhibited hippocampal LTP in a transgenic mouse model also through the production of ROS [69].

### 2.3. Heme

Another important iron containing molecule that has been implicated in the pathogenesis of AD is heme. Heme is an essential molecule in various physiological and pathological mechanisms [70]. It has been suggested that AD patients had lower hemoglobin levels and smaller cell volume with respect to normal aging controls [71]. Complexes II, III, and IV of the electron transport chain require heme to assembly cytochromes needed to function [72]. Perturbation in heme metabolism can cause oxidative stress and cell death [73]. Sankar and collaborators found that heme can mitigate the Aβ 42-mediated neuroinflammation activated by astrocytes [74]. Heme can also bind to Aβ, and this complex is known to have peroxidase activity able to oxidize serotonin and DOPA and providing an intriguing link between heme and oxidative stress in AD [75]. Thus, heme deficiency is followed by formation of APP dimers and loss of complex IV of the electron transport chain, and this event causes OS [76]. This finding suggested that iron accumulation observed in AD could be strictly linked to heme deficiency [76].

### 2.4. Neuroinflammation

Another crucial mechanism through which the presence of Aβ induced oxidative stress is neuroinflammation [77]. Neuroinflammation is considered as an immunological response characterized by the activation of glial cells and the production of inflammatory mediators [78]. Numerous studies revealed a strong correlation between neuroinflammation and AD pathology [79,80,81]; thus, inflammatory cytokines have been reported to increase in the progression of mild cognitive impairment to overt AD [82]. A microarray study of Cribbs and collaborators performed on young, aged, and AD cases demonstrated an upregulation of the innate immune response in aging brains and a slight increase in related genes [83], suggesting that inflammation has a role in the preclinical stages of AD. In this context, microglia play a leading role in neuroinflammation [78]. Aβ can bind different microglial receptors, resulting in the production not only of inflammatory cytokines and chemokines [83] but also of a large amount of oxygen free radicals (^•^OH and O₂^•^) [84], nitric oxide (^•^NO) [85], and tumour necrosis factor (TNF) α [86]. The NLR family pyrin domain containing 3 (NLRP3) inflammasome is a recently found cytoplasmic protein complex involved in neuroinflammation and innate immune response [87]. Recent studies demonstrated that Aβ induce NLRP3 activation in microglia and astrocytes. This event results in the production of caspase 1 and induces the release of cytokines such as IL1β and IL-18, resulting in irreversible damage. On the other hand, the inhibition of NLRP3 inflammasome inhibits Aβ deposition and had a neuroprotective effect in a transgenic AD mouse model [88,89,90,91]. Thus, all these findings suggest that Aβ is a crucial factor in AD associated inflammation and oxidative stress.

### 2.5. NF-kB Pathway

The NF-kB family also has an important role in modulating oxidative stress. Thus, evidence from in vitro studies showed increased oxidative stress-mediated by NF-kB in response to neurons exposed to Aβ; this increased oxidative stress resulted in the accumulation of lipid peroxides and neurodegeneration [92]. Moreover, many studies confirm that Aβ peptides stimulate NF-kB gene expression and its nuclear translocation [93,94,95].

## 3. Oxidative Stress vs. Aβ

As mentioned in the Introduction Section, neurons, unlike other cells, use a large amount of oxygen; therefore, their mitochondria produce large amounts of ATP. This aspect is closely linked to the fact that they are production sites of reactive oxygen species and extremely sensitive to the resulting damage [44].

OS increase is considered to be an early event in AD pathology [96,97], as it contributes to membrane damage, cytoskeleton alterations, and cell death [98].

Thus, the identification of a large number of oxidatively modified proteins in common AD and AD animal models [99,100] suggests that OS plays an important role in AD pathogenesis. Moreover, extensive oxidative damage observed in mild cognitive impairment (MCI) [101] suggests that OS may be an early event in the progression from normal aging to AD pathology. Based on these notions, it seems likely that increased production of ROS may act as important mediators of synaptic loss and eventually promote senile plaques formation [102] (Figure 2).

### 3.1. APP Processing and Secretases

Many authors have confirmed the hypothesis that oxidative stress represents a common mechanism that mediates the accumulation and toxicity of Aβ [103]. Therefore, oxidative stress can be considered one of the factors responsible for the accumulation of Aβ. Oxidizing agents increase the expression of APP [104] and increase the intracellular and secreted Aβ levels [99]. We, along with other authors, have shown that the expression and activity of BACE 1 increased in conditions of oxidative stress [105,106,107,108,109]. Furthermore, there is a significant correlation between Bace1 activity and oxidative stress markers in brain tissues of sporadic AD [110] in which BACE1 levels have also been shown to significantly increased compared to normal aging [111,112].

Many authors proposed a sequence of events that observe oxidative stress, the hyper-expression of BACE1, and the induction of apoptosis connected with the deposition of Aβ. Thus, we have shown that oxidizing agents and HNE increase the expression and activity of BACE1 in neuronally differentiated cells, without modifying the levels of APP [106,109,113]. These events are followed by both Aβ overproduction and apoptosis [114]

Furthermore, it has been shown that oxidative stress increases the activity of gamma secretase both in vitro and in vivo, and the increased expression of BACE1 induced by OS is regulated by gamma secretase [115,116]. These results are of great importance for understanding the pathogenesis of sporadic AD. First of all, it is shown that oxidative stress, which originates during aging, can increase the activity of both secretases (β and γ secretases) and, thus, mediate the hyper production of Aβ. Oxidative stress is the only known factor capable of increasing the cut on APP operated by γ-secretase by increasing the activity of PS1, the catalytic subunit of secretase. These results demonstrate the existence of a vicious circle in which the increased activity of γ-secretase results in an increase in the expression of BACE 1 [117].

Other authors also demonstrated a correlation between the induction of OS and the increase in γ-secretase cleavage on APP [118,119,120,121]. Given that OS can mediate both γ-secretase and BACE1 activities, OS could be considered the molecular link between β-secretase and γ-secretase.

Another important consideration that closely links oxidative stress to the pathogenesis of AD is the fact that many of the best-known risk factors for sporadic AD observe oxidative stress playing a crucial role in their pathogenesis.

### 3.2. Stroke and Hypoxia

It is known in the literature that patients who have had strokes or cerebral infarction have an increased risk of developing AD [122].

The risk becomes even higher in the presence of atherosclerotic risk factors [123]. It has been proposed that hypoxia can alter the metabolism of APP, increasing the activity of β and γ secretase. Sun et al. demonstrated, for the first time, that hypoxia significantly increases the expression and activity of BACE1 [124]. Again, the same authors show that hypoxia increases the deposition of Aβ and the formation of senile plaques, and it also mediates mnemonic deficit, suggesting a mechanism capable of linking vascular damage with AD [125].

Hypoxia is also capable of increasing the activity of γ-secretase; thus, hypoxia inducible factor (HIF)-1α binds the promoter of anteriorpharynx-defectivephenotype (APH-1), a key component of the secretase γ complex, inducing its hyper regulation [125].

Taken together, these data show that hypoxia increases the activity of both secretases, thus resulting in an overproduction of Aβ and the formation of plaques in both in vivo and in vitro models. It is well known that the only effective treatment for cerebral ischemic disease is the rapid restoration of the blood flow to the brain. However, reperfusion often aggravates brain injury, resulting in cerebral ischemia/reperfusion (I/R) injury [126]. The pathophysiological process underlying cerebral I/R injury is complex. Currently, intracellular Ca^2+^ overloading, oxygen free radical injury, excitatory amino acid toxicity, chemokines, and white blood cell interactions are considered to be underlying pathogenic factors that contribute to cerebral I/R injury [127].

It has also been shown that intracellular ROS levels change also during hypoxia; thus, ROS increases during hypoxia [128], and mitochondria would be their source of production under hypoxic conditions [128].

It is accepted that hypoxia increases ROS via the mitochondrial transport chain and specifically by the function of complex III [129].

Mitochondrial-derived ROS appears to be sufficient for stabilizing and activating HIF-1α, and it has been shown that antioxidant compounds can protect against this activation [130]. Recently, some anticancer effects of antioxidant drugs have been precisely ascribed to the protection of events mediated by the activation of HIF-1α [131]. Guglielmotto et al. have shown that hypoxia induces a hyper regulation of BACE1 through two different mechanisms: by inducing an early release of ROS from the mitochondria and through a later activation of HIF-1 [132]. The crucial involvement of ROS released by mitochondria was confirmed by the complete protection exerted by rotenone, which alters the complex 1 of the mitochondrial electron transport chain [132].

### 3.3. Hyperglycemia and AGEs

Diabetes mellitus and hyperglycemia are other risks factor for AD [133].

Hyperglycemia enhances the formation of advanced glycation end products (AGEs), which are senescent protein derivatives from the auto-oxidation of glucose and fructose [134].

Accumulation of AGEs in various tissues is known to occur in normal aging and, at an extremely accelerated rate, in diabetes mellitus and renal failure [134]. AGEs have been detected in vascular walls, lipoproteins, and lipid constituents where they result in macro and microangiopathy and amyloidosis [135].

Increased extracellular AGEs formation was demonstrated in amyloid plaques in different cortical areas [136], where they may have a role in accelerating the conversion of Aβ from monomers to oligomers or higher molecular weight forms.

In addition to post-translational protein modifications, AGEs have other pathologic effects at the cellular and molecular levels.

An increased presence of AGEs has been observed in amyloid plaques in various cerebral cortical areas [137], and their role has been demonstrated in accelerating the conversion of Aβ monomers into high molecular weight oligomers. In addition to inducing post-translational changes, AGEs induce toxicity at the molecular and cellular level through other mechanisms as well. These mechanisms also include the ability to induce oxidative stress, in particular, superoxide and hydrogen peroxide [138,139]. In fact, glycated proteins increase the production of free radicals compared to normal proteins. Another mechanism through which AGEs induce oxidative stress is through binding to their receptor (RAGE), a surface receptor belonging to the immunoglobulin superfamily [140].

RAGE is age-dependent hyper-regulated in human tissues [133], and the increased interaction between AGE and RAGE induces oxidative stress, which is crucial in the pathogenesis of many aging-related diseases, such as diabetes, cardiovascular disease, and AD [141].

The production of ROS induced by the interaction between AGEs and their receptor induces brain damage because it increases the vulnerability of the brain to oxidative stress. ROS increases the synthesis of Aβ, which induces the synthesis of high-mobility group box 1 (HMGB1), and S100. These molecules interact with RAGE, inducing a further production of oxygen free radicals that result in Alzheimer’s disease. Therefore, the induction of oxidative stress precedes the formation of amyloid plaques. In conclusion, we can say that the disease begins with the stress induced by the interaction between AGE and RAGE, and Aβ, HMGB1, and S100 make it progress. The reduction in the levels of AGEs and RAGE, the increase in sRAGE, and the use of antioxidants could really have benefits in the prevention and in slowing down the progression of AD [142,143]. Aβ extracellular fibrillar aggregates have characteristics of AGEs and bind to RAGE in neurons and brain endothelial cells. AGE and Aβ binding to RAGEs results in further oxidative stress that contributes to neuronal death and vascular dementia in AD [144].

Although the brain was once considered an insulin-independent organ, insulin signalling is now recognised as being central to neuronal health and to the function of synapses and brain circuits [145]. Defective brain insulin signalling, as well as related signalling by insulin-like growth factor 1 (IGF-1), is associated with neurological disorders, including Alzheimer’s disease, suggesting that cognitive impairment could be related to a state of brain insulin resistance [146]. Aβ also regulates insulin and IGF signaling in the brain by binding to the insulin receptor and disrupting its signalling capacity in in vivo monkeys injected with oligomeric Aβ [147]. In vitro studies showed that Aβ decreases cell surface insulin receptor expression and promotes synaptic spine loss, one of the earliest structural defects observed in AD. Furthermore, insulin prevents Aβ binding to the synapses preserving synaptic integrity [148]. Taken together, these findings suggest that AD pathological processes may be triggered or exacerbated by peripheral insulin resistance and that Aβ may itself induce brain insulin resistance and synapse loss, raising the possibility that insulin treatment may correct these pathological events. Moreover, IGF-1 protects against hyperglycaemia-induced oxidative stress, and defective insulin signalling makes neurons more vulnerable to oxidative stress [149,150].

The fact that altered glucose metabolism precedes clinical symptoms strongly suggests that perturbed glucose homeostasis may be a cause rather than a consequence of AD. In order to sustain neuroenergetics, brain glucose levels are tightly controlled by the glucose transporter, GLUT1, at the blood–brain barrier (BBB), and its expression is directly correlated with cerebrospinal fluid (CSF) glucose levels, brain development, and function [151,152]. Decreased GLUT1 expression at the BBB in AD brains has been reported [153,154]. Brain glucose enters glycolytic pathways to support its urgent and complex metabolic needs. For example, neurons utilize lactate from astrocytes to support their energy demands. This phenomenon, termed the astrocyte-to-neuron lactate shuttle (ANLS), was first described by Pellerin and Magistretti in the 1990s [155] and is supported by accumulating evidence [156,157]. Notably, increased lactate is associated with tau and Aβ deposition, see review [158]. A human in vivo PET study revealed a spatial correlation between glycolysis and Aβ deposition, suggesting a possible link between region-specific glycolysis in young brains and the subsequent development of AD pathology [159].

### 3.4. Cholesterol

Over the past decade, hypercholesterolemia, characterized by high blood levels of cholesterol, and obesity have been considered risk factors for the development of neurodegenerative diseases [160]. One of the mechanisms proposed to explain this correlation seems to be that high levels of cholesterol and free fatty acids result in incorrect regulation of lipid metabolism, which alters the permeability of BBB inducing neuroinflammation and cognitive deficits [161].

Evidence demonstrates the hypothesis that the oxidation of cholesterol plays a crucial role in the pathogenesis of AD, and that oxysterols represent the link between AD and the alteration of lipid metabolism and hypercholesterolemia. The rationale for this hypothesis lies in the fact that oxysterols, unlike cholesterol, can cross the BBB. The key role of oxysterols in the pathogenesis of AD is suggested by studies that demonstrate their involvement in the modulation of neuroinflammation, in the accumulation of Aβ, and in the induction of cell death [162,163].

## 4. Concluding Remarks

All data summarized in this review postulate that OS and Aβ are linked to each other because Aβ induces OS in vivo and in vitro, and OS increases the production of Aβ; for this reason, the answer to the question “did the chicken come before the egg?” remains, in our opinion, at least in part unsolved. Altogether, the evidence for the primary occurrence of oxidative stress in AD is very attractive and well supported. Zhu and colleagues, many years ago but still very relevant, have proposed a “Two-Hit” hypothesis whereby the early and progressive oxidative damage to neurons elicits a compensatory response such that the cell can exist in the overly oxidizing environment [164]. This “oxidative steady state”, while initially instituted for protection, makes the cell vulnerable to additional insults, such as Aβ deposition, NFT formation, cell cycle aberration, etc. [164]. More importantly, many of the above-mentioned hallmarks, in particular Aβ, are themselves the sources of oxidative stress. The overall effect is positive feedback.

Thus, whether the chicken or the egg comes first, oxidative stress certainly plays a very important role in the pathogenesis of Alzheimer’s disease, and this lays the basis for clinical interventions in AD with antioxidants.

Endogenous antioxidants can protect against the production of free radicals and can mediate the expression of molecules that enhance the antioxidant action in neurons.

Several studies suggested an effectiveness of antioxidant enzymes such as superoxide dismutase (SOD) and catalase (CAT) in slowing the progression of AD. In addition, various dietary antioxidants such as Vitamin E/C, curcumin, resveratrol, and Ginko biloba have emerged as promising agents in preclinical studies. However, the efficacy of antioxidants in clinical practice still leaves open questions. Thus, preclinical intervention with antioxidant molecules could be protective against AD, while their late use is useless [165].

Alzheimer’s disease pathology begins many years before its symptoms; therefore, we must take advantage of this long window to modify and modulate all known risk factors for the disease. In this context, antioxidants can be an important weapon of prevention.

## Figures and Tables

**Figure 1 antioxidants-10-01479-f001:**
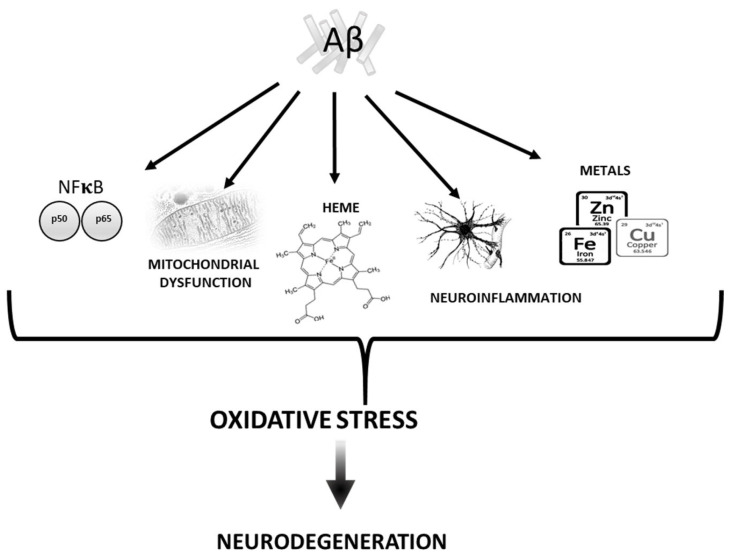
Diagram sketching mechanisms by which Aβ induces oxidative stress.

**Figure 2 antioxidants-10-01479-f002:**
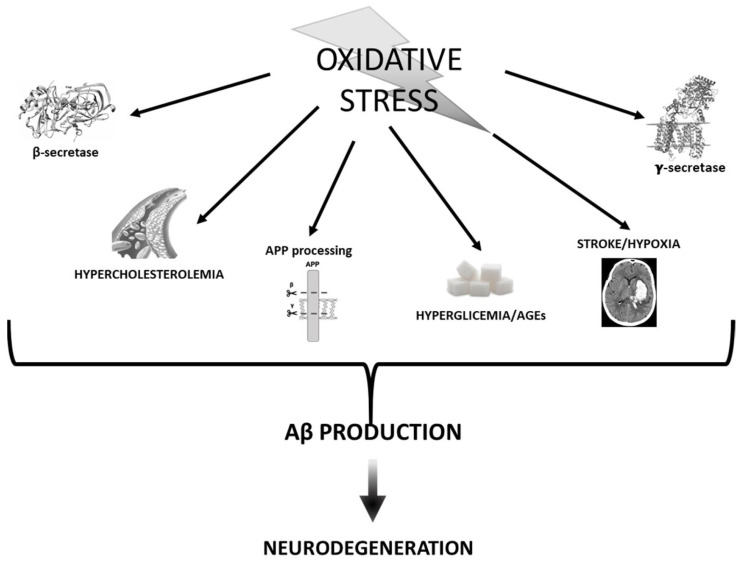
Diagram sketching mechanisms by which oxidative stress mediates Aβ production.

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
