# Peer review of "Oxidative Stress and Beta Amyloid in Alzheimer’s Disease. Which Comes First: The Chicken or the Egg?"

_antioxidants, 2021, doi:10.3390/antiox10091479_

Round 1

Reviewer 1 Report

The article has improved, and some aspects have been sharpened. Still, there are many mistakes (both with regards to the English language and scientifically), several misleading statements, incomplete and incorrect coverage, representation, and citation of literature, etc. The article would benefit from a careful and comprehensive work up for accuracy.

Throughout the revised version, there is now a mix up of references. Many citations are misplaced and not matching the information in the text. That needs to be carefully revised and corrected.

In particular, the paragraphs from lines 67 to 98 should be carefully revised:        

Other than in the previous version of this manuscript, the authors now make a case for different soluble Aβ forms as the main driver of disease onset and progression based on 5 citations from which two papers present original research. Both original articles (Piccini from 2005 and Di Fede from 2018) demonstrate different distributions of soluble Aβ subtypes found in a small amount of postmortem AD human brains; one (Ref 34) study doesn’t include normal control brains, rather compares fAD with sAD. This new angle on Aβ appears like “tweaking the case” by cherry-picking literature. While the cited papers could provide circumstantial evidence for an association of Aβ subtype composition with AD, this is still a hypothetical assumption which has not been adapted by the scientific community at large. In addition, the authors incorrectly represent the studies:

Lines 88-89: “…and it was found that the species of sAβ in the two different conditions are different [33]”. This is a misrepresentation. The most updated original data for this notion are in Ref 35 demonstrating that while both NA and sAD have the same Aβ species, their composition (better would be to say amounts) are different, and that these subtypes cause slightly more neurotoxicity in vitro when derived from sAD brains (this could be due to their different amounts or composition and aggregation).

Lines 89-92: “…than the Aβ species associated with normal brain aging [34]”. There were no normal brains included in the Ref 34 study. This reference is also cited incompletely (the PubMed information is: Sci Rep 2018 Feb 19;8(1):3269)).

Lines 93-94: “…there is a balance between Aβ synthesis and degradation through self-regulatory pathways [3335]”. This has not been analyzed or demonstrated in the cited study. Also, there is a reference missing for the following sentence: “However, the destruction of this equilibrium results in the overproduction of Aβ in brain tissue.”

It is an overstatement and inappropriate to say that “These discoveries have revolutionized the concept of amyloid toxicity and changed the interpretation of pathogenesis.” Do the authors want to imply that the pathogenesis of AD now needs to be rewritten? Given the plethora of literature on Aβ, the authors should, at a minimum, put their argument in perspective with findings from other studies, e.g., a recent paper has not found differences in homomeric, heteromeric, and soluble Aβ between normal and sAD brains (PMID: 31417354).

In my view, there is still too little discussion on metabolism and bioenergetics. Addressing bioenergetics, which is driven by respiration and glycolysis, and many metabolic processes, is essential when discussing ROS.

In the abstract, the authors say “the evidence for the primary occurrence of oxidative stress in AD is attractive”, “evidence indicates that a long period of gradual oxidative damage accumulation precedes and leads to appearance of clinical and pathological AD symptoms including metabolic dysfunction”; “many of the most important risk factors for AD see oxidative stress play an important role”; “Alzheimer's disease begins many years before symptoms”, however, there is very little text addressing exactly these issues.

Reviewer 2 Report

The authors have adequately addressed m previous concerns.

Author Response

We thank's the review

Reviewer 3 Report

The manuscript is a very good review linking the roles of oxidative stress and beta amyloid  in AD pathogenesis. 

The manuscript is well written and organised in sections, properly described. References are adequate and updated.

My only suggestions are as follows:

  • Line 56: remove Ref [26]
  • Line 114: Please spell out all the acronyms in the paper main text such as OXPHOS, as reported above in lines 111-112, which have been removed. 
  • add to ref 72 the following paper relative to key factors sustaining neuroinflammation as a consequence of microglia activation by neurotoxic stimuli. Villa V. et al. Novel celecoxib analogues inhibit glial production of prostaglandinE2, nitric oxide, and oxygen radicals reverting the neuroinflammatoryresponses induced by misfolded prion protein fragment 90-231 orlipopolysaccharide. Pharmacological Research 2016, 113, 500–514
  • line 293-297: remove the related sentences, since they are a repetition of the following ones (see lines 298-302)
  • add to Ref 137 and at the end of line 374 these recent paper, discussing also the antioxidant properties of quinone-based derivatives as anti-AD agents.  Campora, M. et al. Journey on Naphthoquinone and Anthraquinone Derivatives: New Insights in Alzheimer’s Disease. Pharmaceuticals 2021, 14, 33.

I consider this manuscript of interest for readers working in the field, and I suggest its publication after the following minor revisions.

Author Response

This manuscript is a resubmission of an earlier submission. The following is a list of the peer review reports and author responses from that submission.

Round 1

Reviewer 1 Report

This review article summarizes the connection and interaction of oxidative stress (OS) and Ab pathology in Alzheimer’s disease (AD). The authors discuss the question “which comes first: the chicken or the egg?” referring to these two parameters as critical in AD pathogenesis. In their concluding remarks, they infer that, in their opinion, the answer to this question remains unsolved.

Overall, this is a complex research area which has been covered extensively in the literature. With regards to the vast amount of original and review articles on this subject, the current review doesn’t offer much additional novel information or insight. In addition, the subject is brushed over listing a mingle-mangle of information, and partly lacks in depth analysis. The text has several incoherent passages, inconsistencies, and syntactic issues.

Lines 16-18: What do the authors mean by saying “…antioxidant treatment can be an important therapeutic target in order to develop personalized therapies and attacking the disease on several fronts”. There is no discussion about “personalized therapies” in this article.

Lines 28-29 and beyond: The authors say that AD pathology is multifactorial but throughout the text they claim that Ab is the leading cause… It is understood that this review article is about the connection between OS and Ab, however, this is too simplistically represented. sAD pathology is closely linked to age and everything the authors discuss, including Ab accumulation and OS, are part of the (normal) aging process – in fact, age is the greatest risk factor for sAD. The association of Ab with AD is not 1:1, as many people accumulate high levels of Ab plaques but don’t develop dementia and, vice versa, there are people who develop dementia without accumulating over average Ab. The authors assume that Ab is the main driver of AD but in the past years, the field has moved on and this assumption is not that clear anymore. Also, by solely defining AD as a consequence of Ab pathology, the “chicken or egg” question would be obsolete. It rather would be a question of how much OS/Ab action is necessary and at what time in the aging process it would lead to disease development. In other words, the question would refer to the balance between OS and Ab, with Ab may or may not being the driving force in neurodegeneration, at least not in the early phases. Altogether, there is text missing that offers a more distinctive interpretation of mechanisms in AD pathology, including those that are merely reflecting aberrant aging and may (or may not) be associated with AD.

There should be more differentiated text on metabolism and bioenergetics. One of the many hypotheses of AD is that it is a metabolic disorder, and its roots may stem from dysfunctional bioenergetics, insulin/IGF-1 signaling, and other parameters. There is also increasing evidence that metabolic interactions between neurons and other brain cells, e.g., astrocytes, may play a major role in neurodegenerative process. Also, Ab and OS don’t affect neurons alone but also other brain cells. All this should be discussed in more depth.

As a general comment, there should be a careful and clear distinction in data interpretation from studies that claim to model AD and their relevance to human disease. There is a plethora of information from unphysiological study conditions and/or model systems that insufficiently or not at all recapitulate AD revealing only circumstantial evidence for the human disease at best.

Lines 37-43: See comment above, the authors focus on genetics and mention that other factors are involved in the explanation of sAD but then conclude that Ab is the most closely related to its pathogenesis. This is too simplistic and may not be a timely assumption.

Line 69-70: What do the authors mean with saying that Ab undergoes oxidative stress?

Lines 77-80: What is the meaning of neurodevelopment here? Do the authors want to suggest that AD is a developmental disorder?

Lines 156-157: What do the authors mean by saying that “…neurons use high concentrations of oxygen and therefore their mitochondria use large amounts of ATP”? ROS occur in the mitochondria during ATP production … (the following sentence in lines 157-158 is redundant).

Lines 201-203: This sentence is an example of “brushed over” text. There is no substance or detailed information, and the sentence grammatically and syntactically doesn’t make sense.

2.9. Hyperglycemia and AGEs: See comments above. This section is underdeveloped. Hyper- as well as hypoglycemia have been linked to AD and there is a lot more involved in glucose metabolism than just causing AGEs. Also, there is a distinction between insulin/IGF-1 resistance, diabetes, and dysfunctional glycolysis. A good review about this subject is PMID: 30737462.

  1. Concluding Remarks: This section is also very thin and has no real substance. Why do the authors come to the conclusion that the answer remains unsolved? There is no explanation provided. Also, what kind of questions are still open (line 295)? What is the evidence for effectiveness of antioxidants? If the authors want to go down this path, they have to offer much more in-depth information.

There are many misspellings and mistakes that need to be corrected.

Reviewer 2 Report

In the manuscript authored by Tamagno Elena et al, authors tried to answer a very popular question in literature. Despite the topic is interesting, the novelty and originality of this review is limited. Oxidative stress is not adequately described. Nitrosative stress neither. Several reviews can be found considering amyloid beta and oxidative stress and vice-versa as https://doi.org/10.1016/j.mad.2020.111385.

Thus the manuscript in this present form did not add value to the high quality Antioxidant journal 

Reviewer 3 Report

In the manuscript by Elena et al., Authors provide a comprehensive revision of the role of oxidative stress and beta-amyloid in the context of Alzheimer´s disease. The paper is in general well written and suitable for publication in "Antioxidants". Minor revision of English language is necessary.

Line 44 - Authors e supported (something is missing)

Line 46 - Aß is reponsibile for Tau

Line 69 - I don´t see how the peptide Aß undergoes oxidative stress. Maybe authors mean oxidative damage or that the peptide induces/contributes towards oxidative stress.

Line 115 - I don´t understant what authors mean bey heme being a nutrient

Line 139 - Please specify which "oxygen free radicals"

Line 140 - In accordance with IUPAC directives, NO is written with the dot for free radical, before the "N": NO

Line 155 - The abbreviation of versus is vs. (non italic) or vs (italic), both in lower case.

Line 156-157: Neurons do not use a high concentration of O2, but rather a large amount. And mitochondria produce ATP, they do not use it.

Line 163 - what do authors mean by MCI brain regions? and mild cognitive impaiment does nor requir commas

Line 172 - Check numbering, as is skips from 3. to 2.7.

Line 174 - To responsibiliz oxidative stress for accumulation of Aß is quite the overstatement.

Line 223 - I don't think that hypoxia increases ROS production, but rather re-oxygenation. 

Line 293 - Clinical interventions using antioxidants have shown little promise in many other situations, so authors should refer to limitations if they want to "sell" this idea.